# Divergent role of Mitochondrial Amidoxime Reducing Component 1 (MARC1) in human and mouse

**Eriks Smagris**, **Lisa M. Shihanian, Ivory J. Mintah, Parnian Bigdelou, Yuliya Livson, Heather Brown, Niek Verweij, Charleen Hunt, Reid O'Brien Johnson, Tyler J. Greer, Suzanne A. Hartford, George Hindy, Luanluan Sun, Jonas B. Nielsen, Gabor Halasz, Luca A. Lotta, Andrew J. Murphy, Mark W. Sleeman, Viktoria Gusarova***

Regeneron Pharmaceuticals, Tarrytown, New York, Unites States of America

\* Viktoria.Gusarova@regeneron.com

**Data Availability Statement:** All relevant data are within the paper and its Supporting Information files. Gene Expression Omnibus submission is GSE255944, files GSM8083053-GSM8083197.

## Abstract

Recent human genome-wide association studies have identified common missense variants in *MARC1*, p.Ala165Thr and p.Met187Lys, associated with lower hepatic fat, reduction in liver enzymes and protection from most causes of cirrhosis. Using an exome-wide association study we recapitulated earlier *MARC1* p.Ala165Thr and p.Met187Lys findings in 540,000 individuals from five ancestry groups. We also discovered novel rare putative loss of function variants in *MARC1* with a phenotype similar to *MARC1* p.Ala165Thr/p. Met187Lys variants. In vitro studies of recombinant human MARC1 protein revealed Ala165Thr substitution causes protein instability and aberrant localization in hepatic cells, suggesting MARC1 inhibition or deletion may lead to hepatoprotection. Following this hypothesis, we generated *Marc1* knockout mice and evaluated the effect of *Marc1* deletion on liver phenotype. Unexpectedly, our study found that whole-body *Marc1* deficiency in mouse is not protective against hepatic triglyceride accumulation, liver inflammation or fibrosis. In attempts to explain the lack of the observed phenotype, we discovered that Marc1 plays only a minor role in mouse liver while its paralogue Marc2 is the main Marc family enzyme in mice. Our findings highlight the major difference in MARC1 physiological function between human and mouse.

## Author summary

Non-alcoholic fatty liver disease (NAFLD) affects almost a third population worldwide and is the main cause of chronic liver disease and cirrhosis. Recent human genetics studies identified several common variants in *MARC1* gene, associated with lower hepatic fat, reduction in liver enzymes and protection from most causes of cirrhosis, pointing to MARC1 as a potential therapeutic target. Yet, the hepatoprotective mechanism of *MARC1* loss of function is unknown. In this study, we expand human genetic analysis to 540,000 individuals. We confirm previous findings and identify novel rare variants in *MARC1* that similarly associated with reduction in liver fat, liver enzymes and protection from

**Funding:** All authors are employees of Regeneron Pharmaceuticals and all studies in this manuscript are financed by Regeneron Pharmaceuticals. Only authors of this manuscript are contributed to study design, data collection and analysis, decision to publish, or preparation of the manuscript.

**Competing interests:** I have read the journal's policy and the authors of this manuscript have the following competing interests: All authors are employees and shareholders of Regeneron Pharmaceuticals, Inc.

cirrhosis. We also confirm that most common variant in *MARC1* is a true loss of function. We knockdown *Marc1* in mice and challenge them with NAFLD-causing diets to understand its role in liver disease. We find loss of Marc1 in mice did not lead to hepatoprotective effect observed in humans. To understand the discrepancy, we knockdown *Marc1* and its paralogue *Marc2* from primary hepatocytes and discover that Marc1 plays only a minor role in mouse liver while its paralogue Marc2 is the main Marc family enzyme in mice.

## Introduction

Non-alcoholic fatty liver disease (NAFLD) that affects 30% of the US population, is a heterogenous disease characterized by two principal phenotypes: hepatic steatosis, caused by accumulation of excess triglycerides (TGs) in hepatocyte cytosolic lipid droplets and nonalcoholic steatohepatitis (NASH), which is defined by the presence of hepatic steatosis with inflammation, and hepatocellular ballooning, with or without liver fibrosis [1,2]. Despite enormous research effort in this area, the molecular mechanisms of NAFLD are not completely elucidated, making it challenging to develop effective NAFLD therapies. To date, several genome- or exon-wide association studies were conducted to identify genetic variants associated with liver fat accumulation, inflammation and fibrosis. These studies revealed several missense variants that are associated with lipid accumulation and NAFLD progression [3–10]. Recent genome-wide association studies discovered that common *MARC1* missense variants (p. A165T, rs2642438 G>A and p.M187K, rs17850677 T>A) and several rare *MARC1* putative loss of function (pLOF) variants (p.R200Ter, p.R305Ter) in humans are associated with decreased liver fat, lower blood ALT, lower alkaline phosphatase, reduced LDL and total cholesterol, as well as protection from all-cause of liver cirrhosis [11–19]. MARC1 (mitochondrial amidoxime-reducing component 1, known also as MOSC1 and MTARC1) is a molybdenum containing enzyme anchored on the outer mitochondria membrane. MARC1 forms a complex with cytochrome $b_5$ type B (CYB5B) and cytochrome $b_5$ reductase 3 (CYB5R3) and metabolizes a broad range of N-hydroxylated compounds, including amidoximes and N-oxides [20–24]. The MARC1 protein crystal structure revealed it has two MOSC (moco sulfurase C-terminal) domains: MOSC_C domain contains a catalytic site with Mo-molybdopterin cofactor (MOCO) in its center; MOSC_N domain function is unknown, but it is proposed to play role in substrate recognition by binding to MOSC_C domain or by regulating MARC1 interaction with its cofactors CYB5B and CYB5R3 [25]. Intriguingly, both previously-reported MARC1 missense genetic variants, *MARC1* p.A165T and M187K, are located outside of the catalytic domain with the p.A165T variant in an alpha helix of the MOSC_N domain and p.M187K in peptide sequence connecting beta strand and alpha helix in MOSC_N domain [25,26], thus it is unclear how these substitutions contribute to the protective effect against liver disease.

Interestingly, MARC1 closest paralogue MARC2 has a similar protein structure, subcellular distribution and similar, but not identical, preferences to substrates [21,27–29]. Whole body *Marc2* knockout mouse reported to have reduced body weight, resistance to high fat diet induced obesity and protection from fatty liver [30]. Currently, no *Marc1* knockout mice have been described in the literature and although recently published study showed liver-specific siRNA *Marc1* knockdown led to reduction of hepatic triglyceride accumulation [19], it is unclear how *MARC1* pLOF variants affect hepatic metabolism and protect from liver disease in humans.

In this study we used exome-wide association analysis in over 500,000 individuals to identify several novel rare pLOFs variants in *MARC1* with phenotypes similar to previously reported p.Ala165Thr (A165T) and p.Met187Lys (M187K) variants [11–19]. We showed p. A165T substitution in MARC1 causes protein instability, leading to the lower MARC1 protein levels in the livers of human carriers of MARC1 p.A165T variant, confirming this variant is a loss of function. To further understand the mechanism of MARC1 in liver disease, we generated and characterized *Marc1* knockout mice, as well as assessed an enzymatic activity of both Marc1 and Marc2 proteins using CRISPR/Cas9 approach in primary hepatocytes. Our studies highlight the major differences between human and mouse MARC1 function and suggest Marc2 is the major Marc family enzyme in mouse liver.

## Results

### Human *MARC1* variant detection

To expand on *MARC1* genetics and potentially identify novel *MARC1* variants, we performed association testing of genetic variants in the *MARC1* gene with liver and other health-related phenotypes in over 540,000 individuals from five ancestry groups who had available exome sequencing data.

The previously known protein coding variant Ala165Thr (p.A165T) in *MARC1* was significantly associated with 0.50 U/L lower ALT levels (95% CI, -0.45 to -0.56; p = 7.4 x$10^{-75}$) which is similar to individuals carrying rare putative loss of function (pLOF) alleles in *MARC1* (-0.76; 95% CI; -1.25 to -0.27; p = 0.0022; **Fig 1A** and **S1 Table**). In addition, we tested 332 rare (MAF<0.01) missense variants in *MARC1*, of which only Met187Lys (M187K) was significantly associated with lower ALT levels (-0.48 U/L; 95% CI, -0.72 to -0.24; p = 9.9 x $10^{-5}$; **Fig 1A**) [11]. The association of M187K with ALT levels was independent of common fine-mapped variants, including A165T (**S1 Fig**).

In addition to the associations with reduced ALT levels, we found that the burden of rare pLOF variants in MARC1 A165T and M187K were significantly associated with reduced liver fat (**Fig 1B**), as estimated by proton density of the fat fraction which is quantified using magnetic resonance imaging (PDFF-MRI), and with various liver disease etiologies, including alcoholic and non-alcoholic liver diseases, and liver cirrhosis (**Fig 1C** and **S2 Table**). We next estimated the associations of these *MARC1* variants with cardiometabolic traits to better understand the broader impact on health and functional mechanism. We observed significant associations (P<0.001) between *MARC1* loss of function variants and lower circulating apolipoprotein B, LDL-C and HDL-C levels, but higher circulating triglyceride levels (**S2 Fig** and **S3 Table**). In addition, we identified 2 homozygote *MARC1* putative loss of function carriers, one individual in UKB, and one in GHS (**S1 Table**). Both individuals had no reports of liver disease, had normal BMI, average ALT, blood pressure and lipid levels.

### *MARC1* genetic variant p.A165T shows protein instability and aberrant protein localization in hepatocytes

First, we sought to evaluate the effect of *MARC1* pLOF on MARC1 protein expression. Both the published and our own *in silico* prediction studies showed that MARC1 p.A165T can cause protein instability as indicated residue is highly conserved and disrupts the alpha helix [26]. To assess A165T/M187K MARC1 variant effect on expression and localization of MARC1 protein *in vitro*, we overexpressed the human wild-type and two MARC1 variants (A165T and M187K) in a human HuH-7 hepatoma cell line. Levels of MARC1 mRNA were comparable in cells expressing wild-type and the two variants of human MARC1, however the level of A165T

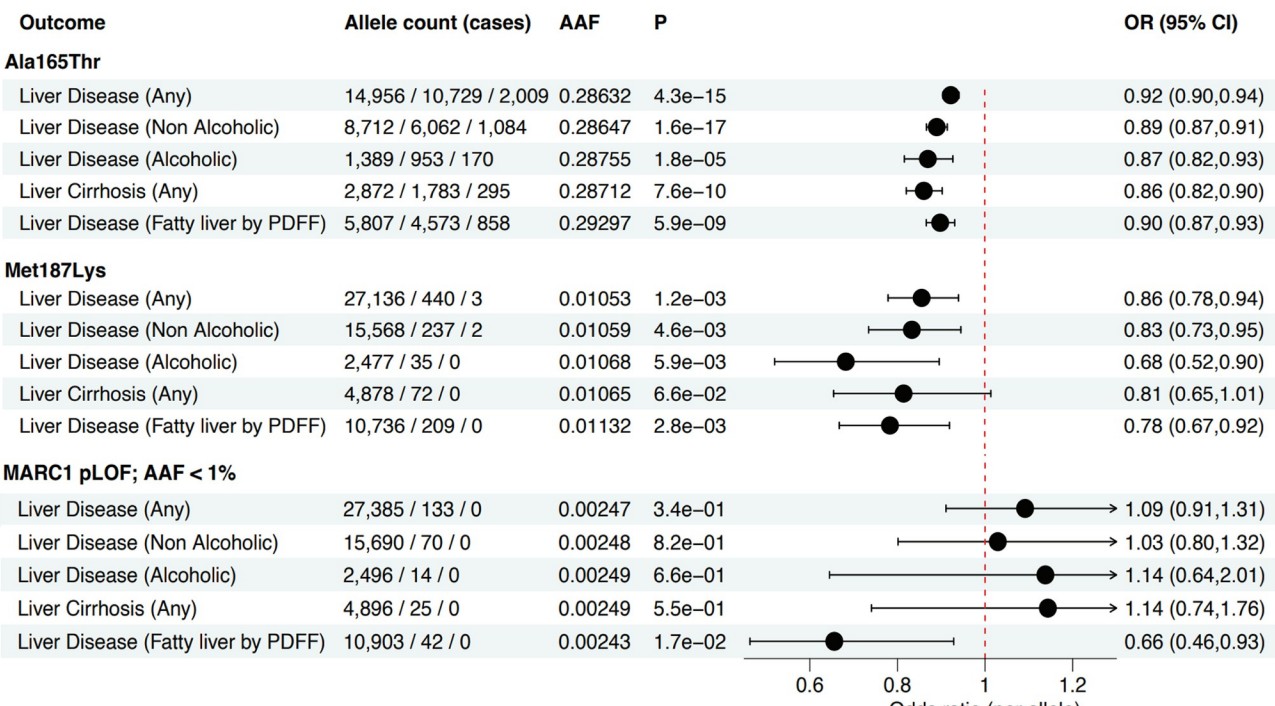

**Fig 1. MARC1 putative loss of function variants (pLOF) show protection from liver disease in exon-wide studies.** (A) *MARC1*, its two common (A165T, M187K) and rare pLOF variants effects on serum ALT, (B) Magnetic resonance imaging (MRI)-estimated proton density liver fat fraction (PDFF) and (C) liver disease (any, liver disease by ICD or by PDFF). Cohort consists of 480,000 controls, except "Fatty liver by PDFF group" which included only 26,000 subjects. ICD—International Classification of Diseases, AAF—alternate allele frequency.

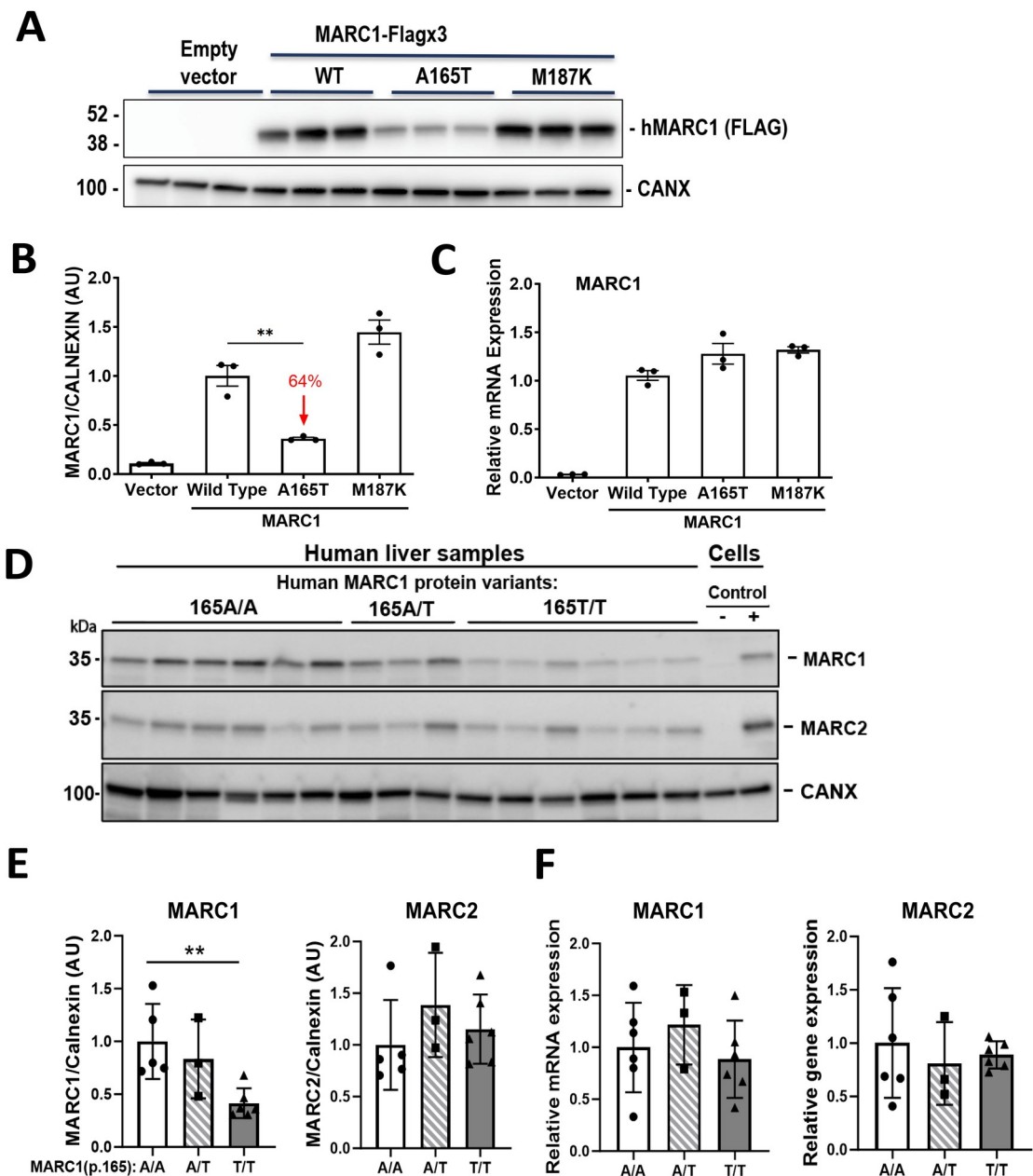

**Fig 2. *MARC1* A165T substitution leads to reduction in MARC1 protein expression.** (A) Plasmids with empty vector (Vector), wild-type and common variants of human MARC1 (pTK promoter, C terminal 3x Flag tag) were expressed in HuH-7 cells. 48 hours after transfection, cells were split into two parts. First part of cells were solubilized in RIPA buffer, proteins separated on SDS-PAGE (4–20%) gels and blotted with corresponding antibodies, (B) immunoblot quantitation was performed. (C) Second part of cells was used to measure *MARC1* mRNA levels using Real-Time PCR. (D) Human liver samples DNA were extracted, and human *MARC1* gene third exon that contains *MARC1* variants p.165A/T and p.187M/K was amplified by PCR and Sanger sequenced, and samples with homozygous *MARC1* p.187M variant selected. Liver samples proteins were extracted, size fractioned and blotted as described in Methods. (E) Western Blot signals were measured with ECL imager build-in software, (F) RNA was extracted from the same livers, and *MARC1*, *MARC2* and house-keeping gene (*36B4*) expression levels were detected by qRT-PCR. For human MARC1 primary antibody control MARC1 deficient HuH-7 cell line (CAS9/Crispr) was used as negative control. For MARC2 primary antibody control MARC1 deficient HuH-7 cell line (expressing no MARC1 and 2) was used as negative control and the same cell line with transient overexpression of recombinant human MARC2 without tags, as positive control. Red arrows–reduction (%) of MARC1 protein. Mean ± s.e.m. are shown in all graphs, **p<0.01.

variant protein was reduced by 64%. Interestingly, MARC1 M187K variant protein levels were similar to MARC1 WT (Fig 2A–2C). Identical experiments with MARC1 variant overexpression were repeated on the second human hepatoma cell line (HepG2) with comparable results (S4 Fig). To avoid a potential non-physiological effect of transfection on protein expression level, we performed human liver protein analysis utilizing livers from A165T variant carriers. Consistent with findings from transient recombinant protein overexpression, the analysis of these human livers revealed 60% reduction in MARC1 p.165T variant protein level compared to the most common (p.165A) variant. Concurrently, the mRNA levels of both variants were similar. No changes were observed in *MARC2* mRNA or protein levels in these same samples (Fig 2D–2F). To uncover the cause of the discrepancy between mRNA and the protein level for A165T variant, we performed confocal microscopy studies in HuH-7 cells. These studies revealed that the protein from cells that overexpressed human recombinant A165T variant showed only partial colocalization within mitochondria, while both proteins from cells overexpressing MARC1 wild-type and M187K variants fully colocalize within mitochondria (Figs 3 and S5). These data suggest that human MARC1 substitution of alanine with threonine in A165T led to partially unstable mislocated protein. To evaluate if A165T protein was degraded by proteasomal or lysosomal degradation, we overexpressed wild-type and A165T variant in HuH-7 cells and incubated cells with or without proteasomal inhibitor MG132 and lysosomal inhibitor chloroquine. The use of either inhibitor did not rescue MARC1 A165T variant protein, (S6A and S6B Fig). In order to differentiate between the compromised translation and stability of the MARC1 protein, we overexpressed wild-type and A165T variant in HuH-7 cells and then treated the cells with the protein synthesis inhibitor cycloheximide (CHX). We also added either MG132 or chloroquine to CHX and compared the expressions of the MARC1 proteins at 0, 3, 6, and 9 hours post-treatment. The addition of MG-132 or chloroquine to CHX for the cells expressing MARC1 p.165T variant did not lead to p.165T variant protein protection from degradation compared to CHX alone treatment, implying that p.165T variant protein instability is not caused by either proteasomal or lysosomal degradation (S6C and S6D Fig).

## Marc1 is predominantly expressed in the liver in mice

In humans, *MARC1* is mainly expressed in white adipose tissue, thyroid, breast and liver, with adipose expression being 3 times higher than the liver (GTEX portal). To estimate tissue distribution and expression levels of *Marc1* mRNA and protein in mouse, we analyzed various tissues from wild type C57BL/6 male mice. *Marc1* mRNA was expressed almost exclusively in the liver, with very minor expression in brown adipose tissues (BAT), as well as in the descending aorta and white adipose tissues (Fig 4A). Similarly, Marc1 protein was mainly expressed in the liver, followed by brown and white adipose tissues (Fig 4C). The closest Marc1 paralogous protein Marc2, was highly expressed in all three parts of the small intestine, liver and kidney consistent with its mRNA expression pattern in mice (Fig 4B and 4C), and with liver being the tissue where human *MARC2* is expressed at the highest level (GTEX portal). Notably, the tissue expression of two cofactors of Marc1 and 2 enzymes (Cyb5b and Cyb5r3) show an expression pattern in mice similar to Marc2 but not Marc1 (Fig 4C).

## Development and validation of *Marc1* knockout mice

To support a hypothesis that A165T of *MARC1* variant is a loss of protein function mutation and to understand if MARC1 deletion or inhibition may have a beneficial effect on liver disease in humans, we generated whole body *Marc1* knockout mouse. The mouse *Marc1* gene

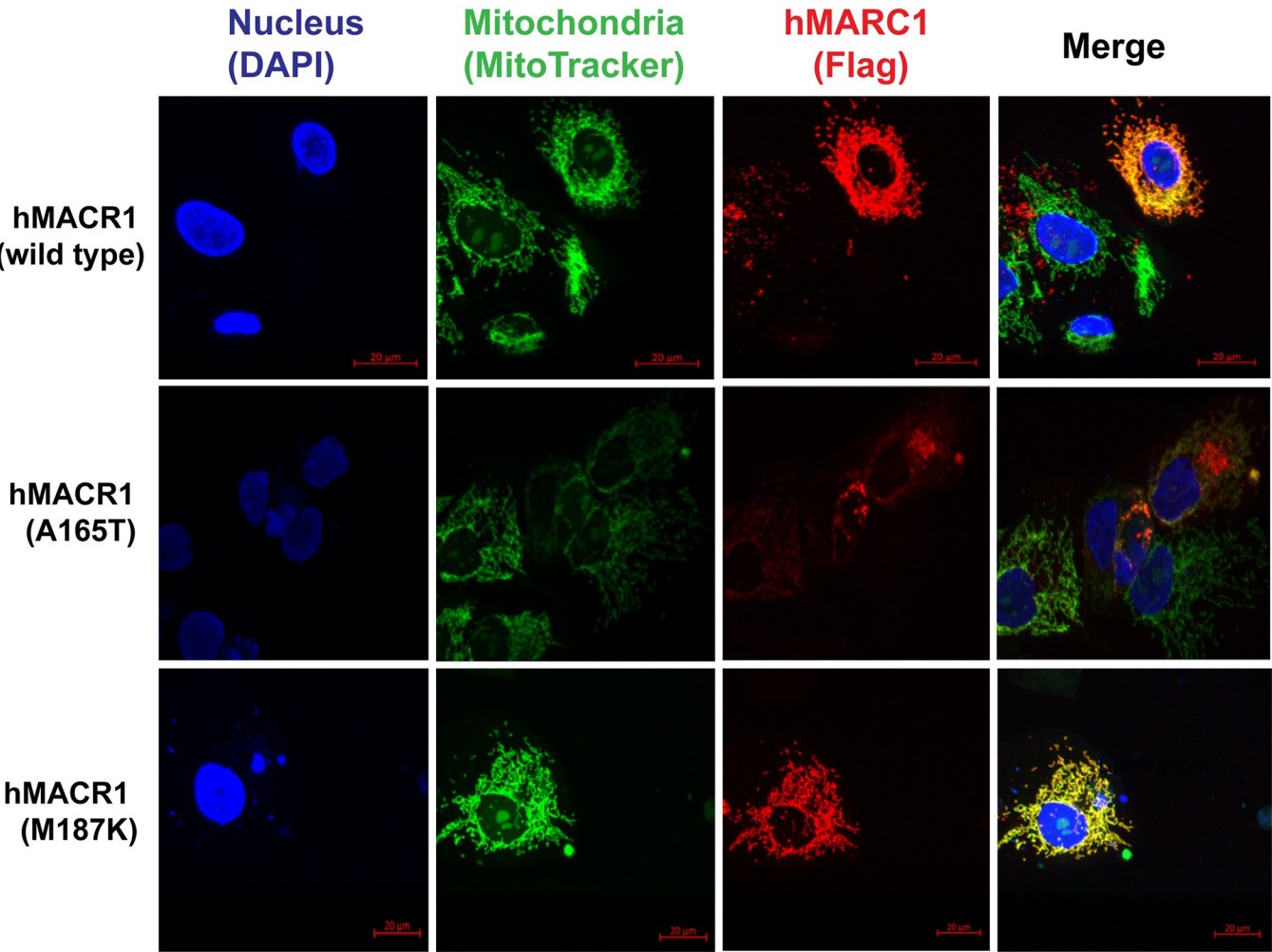

**Fig 3. Human *MARC1* p.165T variant causes mislocalization of MARC1 protein in hepatic cells.** HuH-7 cells were transfected with plasmids containing wild type or 2 common genetic variants (p.A165T and p.M187K) of human MARC1 (with C terminal Flag tag and gene expression driven under CVM promoter). 48 hours after transfection cells were incubated with Mitotracker Orange CMTMRos staining (Thermo Fisher) solution in serum free media for 20 minutes, cells were washed, fixed, permeabilized and blocked as described in Methods. After additional washing, cells were incubated with Flag antibody (Sigma) for 1 hour, washed and incubated with Alexa Fluor 647 conjugated secondary antibody. After final wash, cells were incubated in mounting medium with DAPI (Ibidi) and imaged with Leica confocal microscope. The experiment was repeated once with the equivalent results.

was inactivated by DNA homologous recombination, replacing *Marc1* exon 2 and 3 with antibiotic selection marker, β-galactosidase gene (LacZ), Cre recombinase (self-deleting protamine promoter), as well as with 2 Cre sites. After the first expression of new construct in mouse spermatids antibiotic selection marker and Cre recombinase were deleted by Cre recombination, leaving the β-galactosidase gene (LacZ) and one LoxP site in the flanking intron (**Fig 5A**). The subsequently generated *Marc1*⁻/⁻ (Marc1 KO) mice were born live with the expected Mendelian mode of inheritance and displayed no overtly abnormal phenotype. Adult *Marc1* KO were fertile and nursed their pups normally (**S4 Table**). Full deletion of *Marc1* gene in *Ma*rc1 homozygous KO mouse was confirmed in liver on DNA (PCR), mRNA (qRT-PCR) and protein levels (Western Blot) (**Fig 5B–5D**). Marc2 paralogue or Marc1/2 cofactors (Cyb5B and Cyb5r3) proteins expression were not affected by Marc1 deletion (**Fig 5D**).

## *Marc1* ablation does not elicit hepatoprotective effect in NASH mouse models

Initial phenotyping of *Marc1* KO male mice fed *ad lib* chow diet showed reduction in liver triglycerides and phospholipids, however no changes in serum liver injury markers (ALT and AST), or other markers were observed (S5 Table). Liver gene expression profiling of *Marc1* KO chow fed mice revealed no major changes in gene expression in the liver, including *de novo* lipogenesis or oxidation pathways genes. (S7 and S8 Tables).

To determine Marc1 effect on liver lipid accumulation, inflammation, and fibrosis, we challenged *Marc1* KO and WT male mice with NASH/NAFLD-causing diets: high fat/high fructose diet (HFHFD) and choline-deficient L-amino acid defined high fat diets (CDAA-HFD). Feeding *Marc1* KO and WT mice with HFHFD for 35 weeks did not affect body weight gain, body composition, liver enzymes, and had no effect on serum and liver lipids (Fig 6A–6D and S5 Table). Similarly, there were no significant differences in liver inflammation or fibrosis gene markers between *Marc1* KO and WT mice measured by selected gene expression in the liver by qRT-PCR (Fig 6E). Histological analysis of liver sections from HFHFD fed animals showed no differences between *Marc1* KO and WT male mice in lipid droplet size and distribution (hematoxylin/eosin staining, Fig 7A and 7B). Similarly, we observed no differences between genotypes when quantified hepatic inflammation (CD45 stain) and fibrosis (Sirius red stain) (Fig 7A, 7C and 7D). Feeding *Marc1* KO and WT male mice with the second NASH-causing diet, CDAA-HFD, for 18 weeks did not reveal any significant variation in body weight, body composition or liver lipid accumulation, hepatic inflammation, fibrosis, or any changes in serum lipid levels (S7A–S7E, S8A–S8D Figs and S5 Table). To expand our phenotyping to female mice we utilized the same diets (HFHFD and CDAA-HFD), however found similar lack of phenotype in *Marc1* KO females: no differences in body weight gain, body composition, liver and serum lipids or liver enzymes between female KO and WT mice (S9 Fig and S6 Table).

Focusing only on the effect of *Marc1* deletion on liver steatosis, we fed *Marc1* KO and WT male mice with high sucrose diet (HSD, 9 weeks) or high fat diet (HFD, 11 weeks). However, although we observed significant reduction in ALT levels in *Marc1* KO vs WT in both diets, we did not detect any significant difference in liver triglycerides between WT and *Marc1* KO mice on these diets, consistent with the findings from HFHFD and CDAA-HFD studies described above (S5 Table).

## Marc2 but not Marc1 is a main enzyme with N-hydroxylated amidine reduction properties in mouse liver

To assess Marc1 and 2 enzymatic activities for reduction of N-hydroxyl substrates, we adapted an *in vitro* system utilizing a commonly used synthetic substrate, benzamidoxime (BAO), to evaluate its reduction to benzamidine (BA) [28]. Our initial attempts to use human hepatic or other cell lines (HepG2, ZR-75-1 and HEK293) with knockout or overexpression of human *MARC1* gene described earlier [21,27,28,31] did not show any reliable and consistent measurable BAO reduction and BA production. Thus, we decided to utilize freshly prepared primary mouse hepatocytes to assess N-hydroxyl substrate reduction activity by Marc1 and Marc2 enzymes directly. Our early experiments with plated wild type mouse primary hepatocytes showed BAO reduction was detectable in 2–4 hours' time period, reaching up to 30% conversion of BAO to BA in the media. However, no difference was observed in BAO reduction rates between primary hepatocytes isolated from *Marc1* KO or WT mice. Thus, we hypothesized that the conversion of BAO to BA observed in primary hepatocytes may be driven by Marc1 paralogue, Marc2, which is present in both *Marc1* KO and WT hepatocytes. To evaluate this

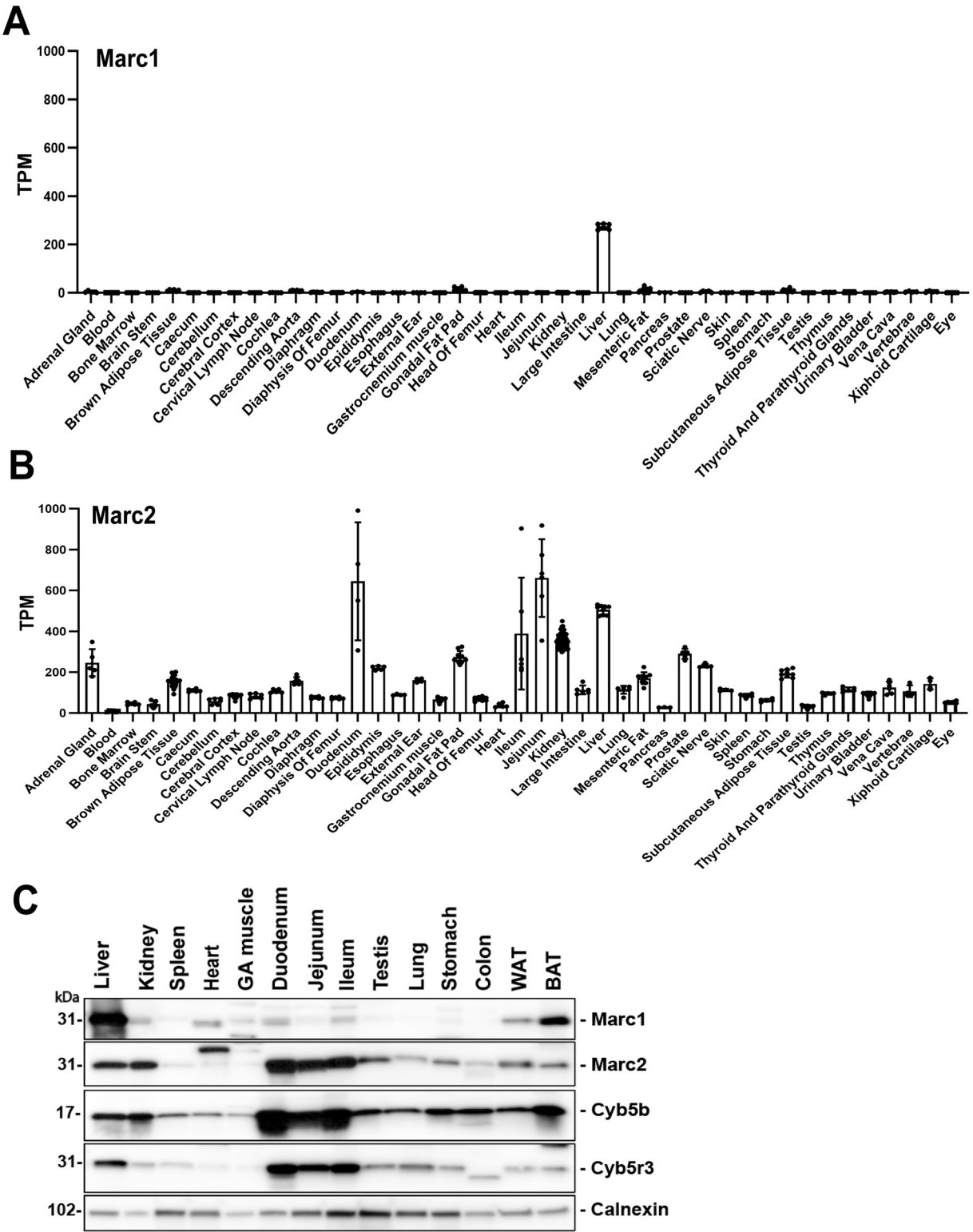

**Fig 4. *Marc1* and *Marc2* have distinct mRNA and protein expression patterns in mouse tissues.** Chow fed 8–10 weeks old wild type C57BL/6N male mice (n = 10) were sacrificed at *ad lib* fed state, tissues collected, and *Marc1* (A) and *Marc2* (B) mRNA gene expression detected by RNAseq as described in Methods. Data plotted as mean with standard error. From some tissues only 3–6 samples were analyzed. Values were calculated as Transcripts Per Million (TPM). (C) 12 weeks old wild type C57BL/6N male mice fed *ad lib* chow diet were sacrificed and represented tissues collected. Tissue proteins were extracted, 20 µg of total protein of each organ were size fractionated by SDS-PAGE (4–20%)

and blotted as described in Methods. Calnexin served as protein loading control for the experiment. GA–Gastrocnemius muscule; WAT–epididymal white adipose tissue; BAT–brown adipose tissue. Small intestine was cut into 3 equal parts, washed and protein extracted.

hypothesis, we partially deleted *Marc1*, *Marc2* or both *Marc1/2* from primary hepatocytes using CRISPR/Cas9 method. The deletion efficiency was evaluated by the respective Western Blot protein analysis (**Fig 8A**). Consistent with previous data, we observed that deletion of *Marc1* gene alone did not affect BAO N-hydroxyl reduction compared with control (anti RFP gRNA) treated hepatocytes. Remarkably, *Marc2* deletion decreased BAO N-hydroxyl reduction by more than 58% and double deletion of *Marc1/2* abolished BAO reduction to BA activity by 72%, compared with control-treated hepatocytes (**Fig 8B**). Residual N-hydroxyl reduction activity (28%) in *Marc1/2* double gene deletion group can be explain with partial

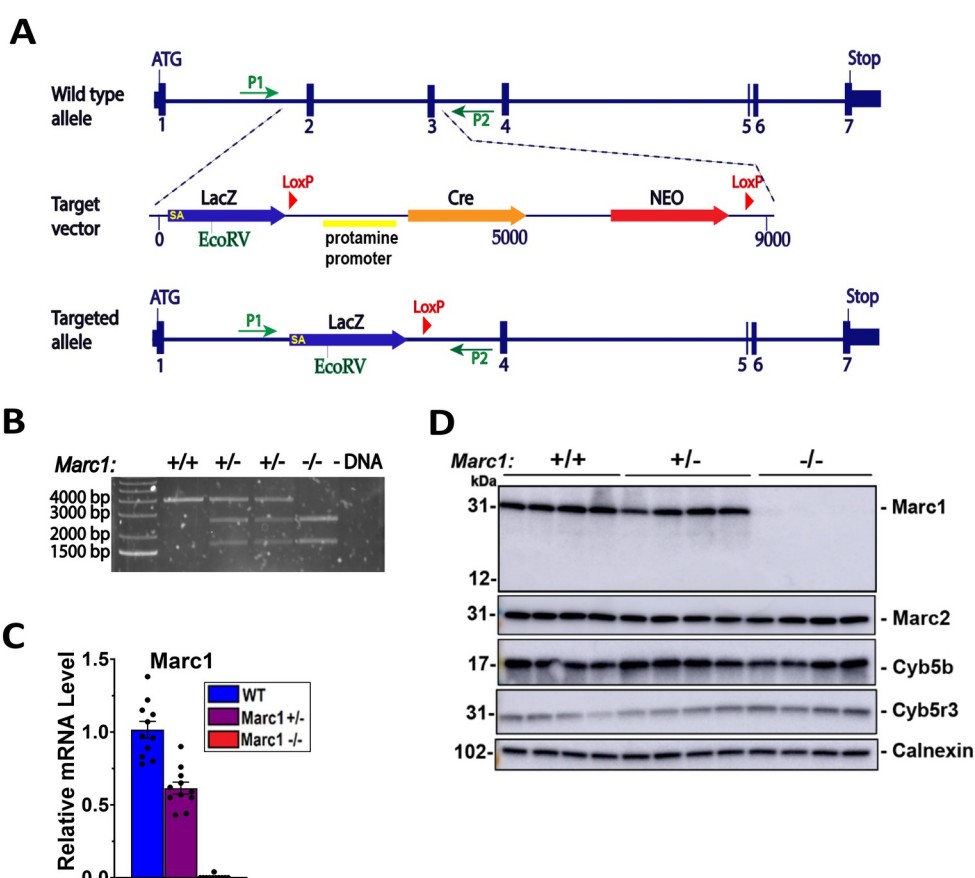

**Fig 5. Generation and validation of *Marc1* knockout mice.** (A) Homologous recombination was used to replace the second and third exons of mouse *Marc1* gene. The antibiotic selection marker (Neo) located between two LoxP sites was deleted with Cre recombinase under self-deleting protamine promoter, leaving β-galactosidase gene (LacZ) and one LoxP site in the flanking intron. (B) Total DNA was extracted from the wild type, heterozygous and homozygous *Marc1* knockout male mice livers using commercial kit. Genotyping was performed by PCR with dedicated primers (green arrows) to amplify a 4 kB fragment from genomic DNA flanking *Marc1* gene second and third exons. Amplified PCR DNA fragments were digested by EcoRV restriction enzyme (present in LacZ, but not in mouse genome form *Marc1* Exon 2 to 3) and size separated on agarose gel by electrophoresis. (C) mRNA (qRT-PCR) and (D) immunoblotting analysis of hepatic *Marc1*, *Marc2* and cofactor proteins in 16-week-old male WT, *Marc1* [+/-] and *Marc1* [-/-] mice. Liver lysates were prepared for Western Blot as described under "Methods". Each sample (45 μg) was size-fractionated by SDS-PAGE (4–15%), and immunoblotting was performed using a rabbit anti-mouse Marc1 polyclonal antibody (DB0161 (1ug/ml); antibody targeting C-terminus of mouse Marc1 protein). Calnexin served as protein loading control for the experiment. PCR–polymerase chain reaction. SA—splice site acceptor.

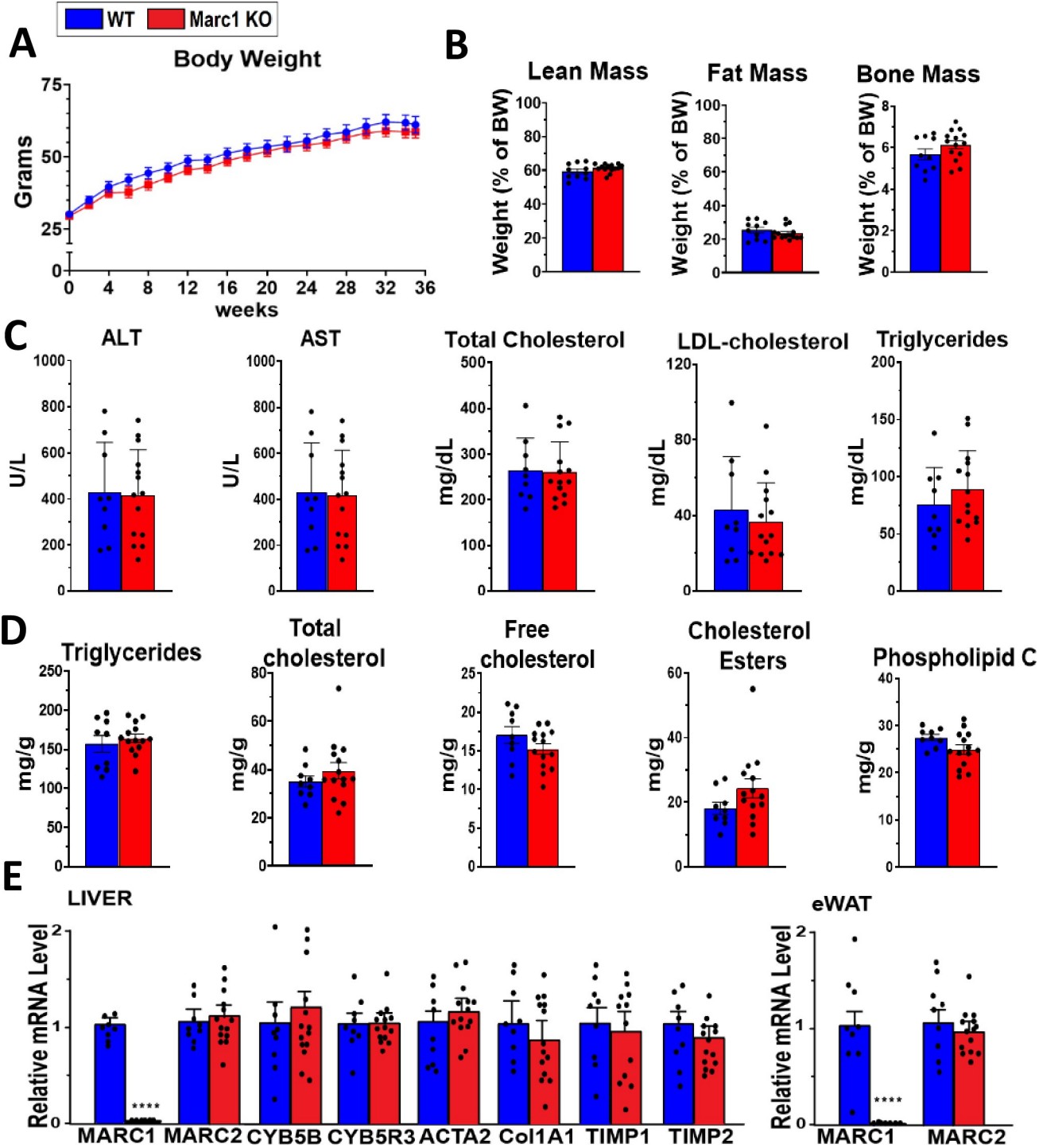

**Fig 6. *Marc1* gene deletion has no effect on circulating or liver lipids in male mice fed High fat high fructose diet (HFHFD).** Eight weeks old male mice (10 WT and 14 Marc1 KO) were fed with HFHFD for 34 weeks. (A) Body mass gain was measured every 2 weeks, (B) body composition (Micro CT) in mice were assessed at week 13 of the experiment. (C) Serum, (D) liver lipids, (E) selected liver gene expression (mRNA by qRT-PCR) were measured at the end of experiment. Each point represents individual mouse, mean ± s.e.m. are shown in all graphs. ****$p < 0.0001$.

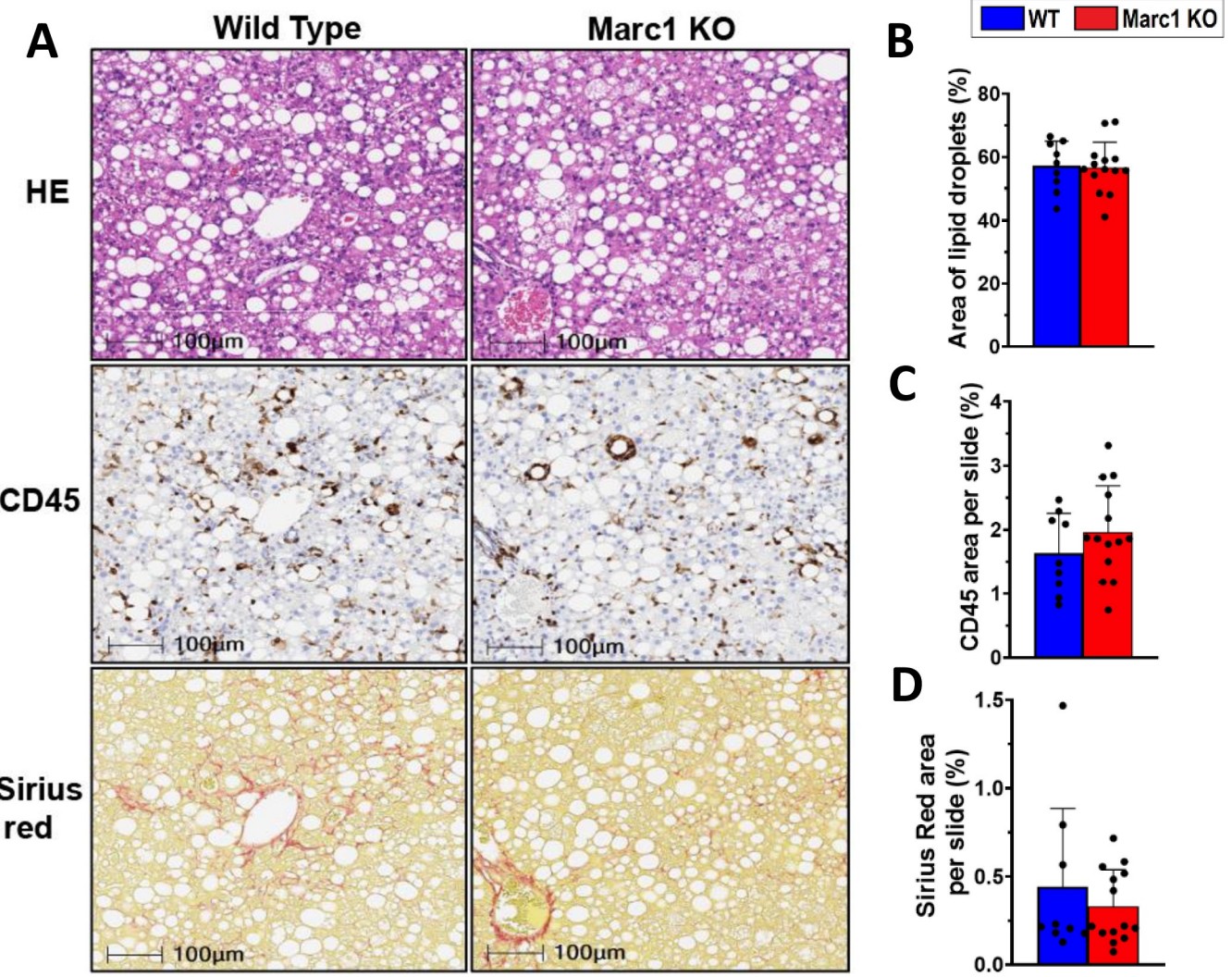

**Fig 7. *Marc1* gene deletion has no effect on liver lipid accumulation, inflammation or fibrosis in mice fed High fat high fructose diet (HFHFD).** (A) Liver samples were collected from mice described in Fig 6. at the end of the experiment. Fixed liver sections were stained with hematoxylin/eosin (HE), CD45 and Sirius Red staining. (B) Lipid droplet area, (C) CD45 positive cell area and (D) fibrosis area were measured from each animal liver section. Slides were scanned on Aperio AT2 scanners (Leica Biosystems) with 10X magnification. Each point represents individual mouse, mean ± s.e.m. are shown in all graphs.

*Marc1/2* gene editing (inactivation) as CRISPR/Cas9 editing happens independently in each hepatocyte; we cannot also exclude that mouse hepatocytes may possess other enzymes with low affinity for BAO reduction. It is worth noting that CRISPR/Cas9 editing could have off-target effect even after careful gRNA selection [32].

## Marc2 knockout mouse develop neurological defects

Marc1 and Marc2 proteins are highly similar (59%) and share the same substrate preference as mentioned earlier. While our human genetic analysis did not suggest MARC2 plays a role in liver disease in humans (**S3 Fig**), we decided to generate *Marc2* KO mice in the attempt to utilize *Marc2* deletion as a surrogate that can help to shed light on the role of molybdenum-containing enzymes on liver disease-relevant conditions in mice. *Marc2* KO mice were generated on 100% C57BL/6 genetic background, using the same technique of gene editing as was used

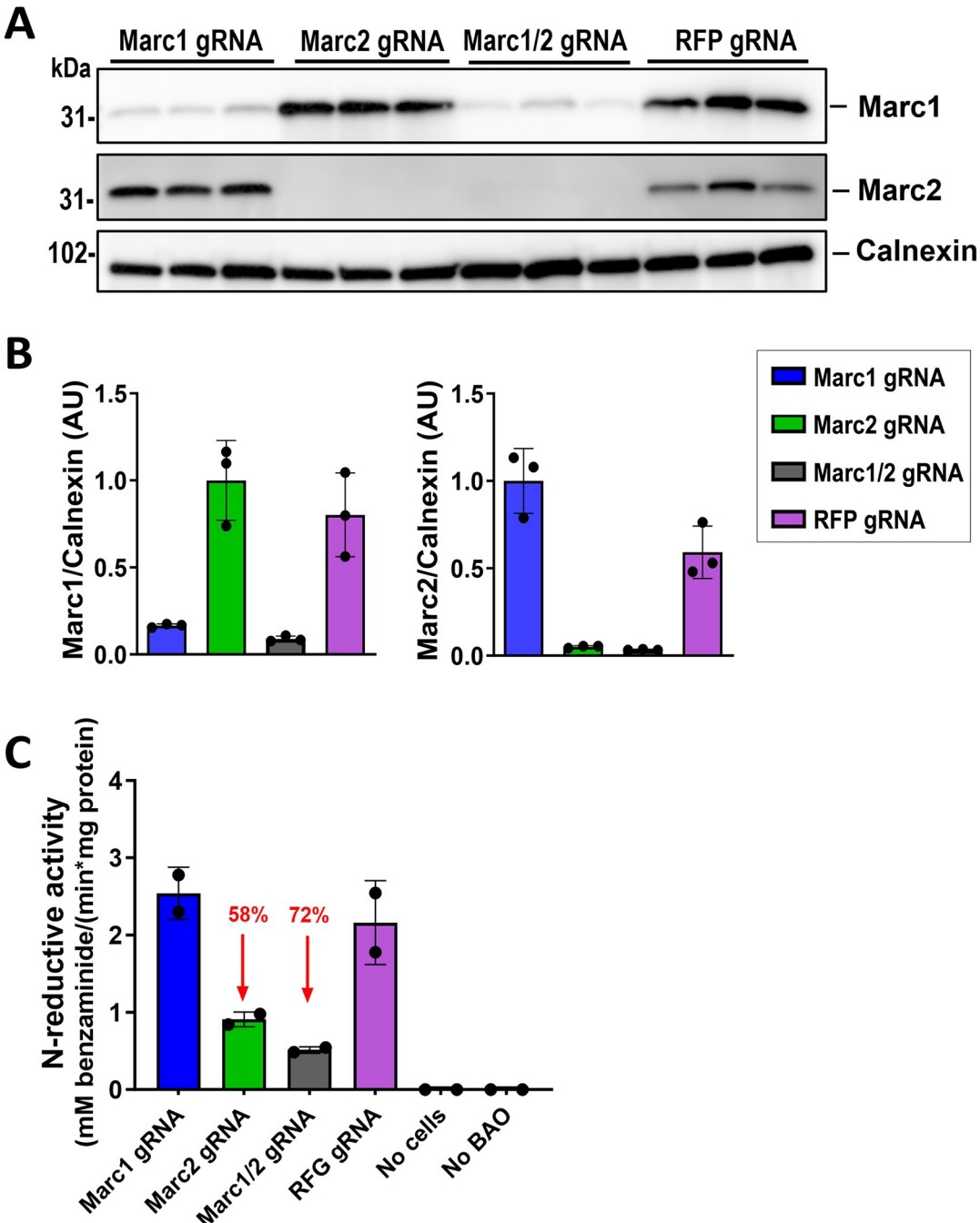

**Fig 8. Marc2 is the main benzamidoxime (BAO) reducing enzyme in mouse hepatocytes.** 12 weeks old male Cas9 transgenic mice were injected in tail vein with $3\times10^{10}$ genome copies (GC) of AAV8 containing Cas9 gRNA against *Marc1*, *Marc2*, *Marc1/2* (edit both genes), and control (red fluorescent protein (*RGF*)). (A) 3 weeks after virus injection mice were sacrificed and primary hepatocytes isolated, counted and plated on poly-D-lysine pre-coated plates, then cells were left in incubator to attach to plate for 4 hours. After attachment, cells were washed with PBS once and incubated in serum free media with 0.5 mM benzamidoxime for 2 hours, then cell media were collected and protein from all primary cell plates was extracted, size fractioned on 4–20% SDS-PAGE gel and blotted with corresponding antibodies. Densitometry was used to evaluate protein reduction after *Marc1* or *Marc2* gene editing and normalized against house-keeping gene (calnexin). (B) BAO and its metabolite benzamidine (BA) were extracted from the collected cell media and measured by UV-HPLC. Controls: no cells–no primary hepatocytes plated for incubation of 0.5 mM BAO in serum free media for 2 hours (control for spontaneous conversion from BAO to BA); no BAO–no benzamidoxime added to RFP gRNA treated cells with serum free media (control for BAO/BA in media or cells). Experiment repeated once with comparable results. Red arrows–% reduction of BAO conversion.

for *Marc1* KO mouse. Unexpectedly, at the age of 7–10 weeks 100% of both sexes of homozygous *Marc2* knockout mice showed rapidly progressing hind limb paralysis. Movement disabilities significantly affected the possibility of the future phenotyping of these mice on NAFLD/NASH-causing diets. Limited characterization of the young mice on chow diet showed reduced body weights and decreased fat mass in *Marc2* KO female mice compared to wild-type (**S10 Fig**), consistent with the previously reported findings [30]. *Marc2* KO mouse liver RNAseq profiling showed a very significant increase of xenobiotic-metabolizing enzymes (such as Fmo3, Cyb2b9, and others), which is consistent with its reported role in detoxification pathways [33]; this signature was not present in *Marc1* KO mouse (**S8 Table**). The cause of *Marc2* KO neurological phenotype is unknown at present as our human genetics analysis did not reveal any association between human *MARC2* pLOFs alleles carriers and neurological defects.

## Discussion

Recent advances in human genetics have led to identification of several genes of interest that may serve as potential therapeutic targets for non-alcoholic liver disease. MARC1 represents one of them, where newly published loss-of-function mutations in *MARC1* were shown to be protective against liver diseases, including most causes of liver cirrhosis [11,17]. In this study, we replicated and expanded on *MARC1* genetic findings by conducting whole exome sequencing analysis in over 540,000 individuals from five ancestry groups. Our study identified several novel rare, putative loss of function variants in *MARC1* gene and showed that these variants have a phenotype similar to the previously reported common variant p.A165T and are associated with protection from liver disease. Additionally, we found that the most common *MARC1* gene variant, p.A165T, causes loss of function of MARC1 protein due to its aberrant subcellular localization and degradation. We generated whole body *Marc1* knockout mice in order to understand the mechanism of the potential hepatoprotective effect of *MARC1* deletion. However, our study found that whole-body *Marc1* deficiency in mouse is not protective against hepatic triglyceride accumulation, inflammation or fibrosis under different dietary challenges that are known to induce these conditions. In our attempts to explain the lack of observed phenotype that contradicts the findings from human genetic studies, we discovered Marc1 plays only a minor role in mouse liver, while its paralogue, Marc2, is the main Marc family enzyme in mice, once again stressing the differences between human and mouse biology.

Mechanistically, it is unknown how *MARC1* pLOFs affects liver disease in humans. Previous studies, using recombinant proteins isolated from *E. coli*, reported A165T variant does not cause difference in N-reduction of benzamidoxime compared to p.165A variant [33,34]. Our current assessment of the most common MARC1 p.A165T variant in hepatic cells revealed that it produces partially mislocalized and unstable protein compared to the MARC1 wild type. This finding confirms that *MARC1* p.A165T variant is a loss of function. Previous human MARC1 protein crystal structure studies indicated that *MARC1* p.A165T variant is located in the conserved MOSC_N domain alpha helix, which lies far away from the enzyme catalytic site in MOSC_C domain [25]. Thus, the A165T substitution is unlikely to affect the catalytic activity of the enzyme directly, but rather diminishes its activity through reduction in hepatic expression due to the disturbed protein intracellular localization and instability. Inconsistently with this finding, the results from overexpression of another *MARC1* pLOF variant p.M187K in liver cell lines did not show changes in protein level or its colocalization within mitochondria (**Figs 2A, 2B, 3, S4 and S5**). While this result is puzzling and may suggest A165T and M187K affect the enzymatic activity of MARC1 differently, it was challenging to

assess due to the luck of reliable *in vitro* or *in vivo* systems. We can only conclude that overexpression of M187K variant in cell lines may not fully reproduce physiological conditions. Unfortunately, due to rareness of M187K variant, it was not feasible to confirm its mRNA and protein levels in human liver as it was performed for A165T variant (**Fig 2D and 2E**).

While multiple genes identified through human genetic analysis shown strong association with liver disease, their mechanistic studies in mouse models proven to be challenging. Current NASH/NAFLD mouse models are far from an ideal instrument for these studies due to disease progression and pathology being different from humans, differences in tissue-specific expression between mice and humans, as well as potential lack of essential co-factors or intermediates. Thus, knockout mice of recently discovered protein-truncating variant in HSD17B13, that leads to generation of unstable protein (essentially knockout) and associated with protection from liver disease in humans, did not show any protective phenotype in mice, or even led to unexpected worsening of the phenotype [35,36]. Original studies in *Pnpla3* knockout mice of genetically validated high risk-NAFLD associated allele I148M, led to enzymatic loss of function of PNPLA3 protein, but showed no phenotype in mouse [37,38]. The follow-up studies discovered that I148M substitution leads to excessive accumulation of the protein on lipid droplets—a new, modified function of the protein leading to the major phenotype difference between KO (lack of phenotype) and I148M knock-in mice (increase in fat accumulation in the liver) [39]. It is noteworthy, that even though I148M has a fat-accumulation phenotype in mouse liver, this phenotype is minor compared to the one observed in humans. This discrepancy is driven by multiple factors with one of them being the difference in mouse vs human expression of *PNPLA3*: the expression of *PNPLA3* is high in human liver but low in adipose tissue, while contrarily it is very low in mouse liver and high in mouse adipose tissue [40,41].

Nevertheless, in the effort to understand the mechanism of how *MARC1* pLOF leads to protection from liver disease, we generated *Marc1* knockout mice and challenged them using the most validated diets, HFHFD and CDAA-HFD, that characterized by the development of all three major stages of NAFLD: steatosis, inflammation and fibrosis. Our attempt to understand how reduction in Marc1 enzyme can lead to protection from liver disease had some limitations. Specifically, *MARC1* mRNA tissue-specific expression comparison between mouse and human revealed that in mouse, *Marc1* mRNA is mostly expressed in liver, followed by very minor expression in brown and white adipose tissues (**Fig 4**), where human *MARC1* mRNA is expressed at highest levels in adipose, with liver expression almost three-fold lower than in adipose tissue (GTEX Portal). Thus, liver is not the main organ expressing *MARC1* in humans, and it could not be excluded that *MARC1* pLOF hepatoprotective effect is mediated by adipose tissues, and not by the liver. Intriguingly, a recently published study utilizing siRNA for reduction of *Marc1* expression in mouse liver suggested that liver-specific deletion of *Marc1* is sufficient to see the hepatoprotective phenotype in rodents, however this study did not utilize nonspecific siRNA control [19]. Our data does not support those findings, as we did not observe any hepatoprotective phenotype in both sexes of whole body homozygous *Marc1* KO mice using multiple NASH/NAFLD diets (**S5 and S6 Tables**).

Conspicuously, our studies in mouse primary hepatocytes using CRISPR/Cas9 deletion of *Marc1*, *Marc2* or *Marc1/2* genes showed that in the mouse liver, Marc2 is the main enzyme responsible for N-reductive biotransformation of N-hydroxylated substrates, which is congruent with the reported data showing major reduction in N-reductive activity in *Marc2* KO mouse liver lysates [30]. Unfortunately, we have not been able to evaluate the effect of *Marc2* deletion on a liver disease-relevant phenotype, due to the severe neurological defect observed in these mice. Mild neurological symptoms (decreased prepulse inhibition and increased startle reflex) were also recently described by European Conditional Mouse Mutagenesis Program

(EUCOMM/IMPC) (https://www.mousephenotype.org/data/genes/MGI:1914497). The over-all phenotype for this mouse was consistent with a previous report by Clement's group [30]; however Clement's paper does not report any neurological findings. The discrepancy between our data and the published studies is unclear, but the differences could be driven by alternative genetic deletion strategies of *Marc2* gene, as well as differences in diet, housing condition or other factors.

In conclusion, our study identified several novel pLOFs in *MARC1* that are associated with reduced liver fat and liver enzymes, consistent with previously reported common variant p. A165T. Our work revealed MARC1 A165T substitution leads to protein degradation and mis-localization, confirming that pLOF A165T is a genuine loss of protein function in humans. While our studies pinpoint Marc1 paralogue Marc2 as a main molybdenum-containing enzyme with N-hydroxylated amidine compounds reducing properties in mouse liver, the lack of a phenotype in *Marc1* KO mice highlights the significant differences in human and mouse MARC1 protein functions, as well as shows challenges in transition from human genetic find-ings to deeper mechanistic studies of MARC1 protein in animal models.

## Material and methods

### Ethics statement

Ethical approval for UK Biobank (UKB) was provided by the North West Multi-center Research Ethics Committee (MREC; 11/NW/0382), which covers the UK. The GHS MyCode study (GHS) is a health system-based cohort of patients from Central and Eastern Pennsylva-nia (USA) recruited in 2007–2019. GHS was approved by the Geisinger Institutional Review Board (2006–0258). All used cohorts patients/participants provided their written informed consent to participate in this study.

### DNA sequencing and genotyping data

High-coverage whole-exome sequencing was performed using NimbleGen VCRome probes or a modified version of the xGen design from IDT. Sequencing was done using Illumina v4 HiSeq 2500 or NovaSeq instruments, achieving over 20x coverage for 96% of VCRome samples and 99% of IDT samples. Variants were annotated using snpEff and Ensembl v85 gene defini-tions, prioritizing protein-coding transcripts based on functional impact. pLOF and missense variants were classified for functional impact according to *in silico* prediction algorithms. Common variant genotyping was performed on one of four SNP array types: Illumina OmniExpress Exome array (OMNI; 59345 samples from GHS), Illumina Global Screening Array (GSA; UPENN-PMBB, BioMe and 82,527 samples from GHS), Applied Biosystems UK BiLEVE Axiom Array (49,950 samples from UK Biobank (UKB)) or Applied Biosystems UK Biobank Axiom Array (438,427 samples from UKB). We retained variants with a minor allele frequency (MAF) >1%, <10% missingness, Hardy-Weinberg equilibrium test P-value>$10^{-15}$. We imputed the genotyped variants using the TOPMed reference panel [42] using the TOPMed imputation server [43]. In GHS, imputation was performed separately by genotyping platform. Further details are provided elsewhere [10].

Genetic analyses were stratified by ancestry and adjusted for age, age2, sex, age-by-sex and age2-by-sex interaction terms, experimental batch-related covariates, the first 10 common var-iant-derived principal components, the first 20 rare variant-derived principal components using REGENIE. Results were combined across subsets by fixed-effect inverse variance-weighted meta-analysis [10].

## Human phenotype definitions

The data cleaning process for continuous traits involved removing non-physiologic lab values and results stemming from invalid or contaminated specimens. The median transaminase values for each individual were extracted from electronic health records at Geisinger Health System (GHS). In the UK Biobank (UKB), transaminases were measured using a Beckman Coulter AU5800 at the baseline study visit. Cases of binary diseases were defined based on one or more of the following criteria: self-reported disease obtained from digital questionnaire or interview with a trained nurse, in-patient hospitalization for the disease, or clinical-problem list entries of the disease, according to International classification of Diseases, Ninth (ICD-9) or Tenth (ICD-10) Revision diagnosis code, medical procedures, or surgery due to the disease, death due to the disease, and a disease diagnosis code entered for two or more outpatient visits in separate calendar days. Controls were defined as individuals who did not meet any of the criteria for case status. To minimize misclassification, the liver disease control group was excluded from analysis if (i) non-cases were diagnosed with any type of liver disease (not restricted to the type of liver disease in question), (ii) non-cases with only one out-patient encounter related to the type of liver disease in question, (iii) non-cases diagnosed with ascites presumably related to liver failure, and (iv) non-cases with elevated alanine aminotransferase (ALT) levels (>33 IU/L for men, >25 IU/L for women). For more details on the specific entries and population characteristics, please refer to [10].

## Mice and procedures

$Marc1^{-/-}$ mice (ID #20561) were generated using Regeneron's VelociGene technology [44,45]. All mice were maintained in the Regeneron Animal Facility during the entire study period. Wild-type (Marc1+/+) and heterozygous (Marc1+/-) littermates were used as controls. All experiments were performed on mixed background mice (75% C57BL/6NTac 25% 129S6/SvEvTac), except RNAseq profiling where mice on 100% C57BL/6NTac background were used.

Marc2 knockout mice (ID #20646) were generated similarly as Marc1 KO mouse using Regeneron VelociGene technology, where second exon of Marc2 gene was replaced with LacZ reporter. Marc2 mice were bred on 100% C57BL/6NTac background.

Cas9-Ready (ver. 2.5, ID #2673) mice, expressing Cas9 transgene under CAG promoter, were generated as previously described [44]. Liver-specific Marc1, Marc2 or Marc1/2 gene ablation was achieved by hepatic Adeno-associated virus serotype 8 (AAV8) delivery ($3x10^{10}$ genome copies (GC) per mouse i/v in tail vein) of CAS9 guide RNA (gRNA) against mouse Marc1 or Marc2 gene exons–for Marc1 only (5′–CTGGCCCGGGACCCGAAACG–3′), for Marc2 only (5′– CCCGATCAAGTCCTGCAAGG–3′) and for Marc1/Marc2 double ablation (5′–ACGCCGAGCCAGGTGGACCG–3′). Cas9 gRNA against red fluorescent protein (RGF) was used as a control. Three weeks after AAV8 injection—ablation of hepatic gene was confirmed by qRT-PCR, next generation sequencing and Western Blot.

Male and female mice (one to five per cage) were maintained on a 12 h light/dark cycle at 22 ± 1°C and were fed ad libitum chow (LabDiet, Cat. No 5053). In some experiments, mice were fed high fat high fructose diet (Research Diets, Cat. No D09100310), choline-deficient L-amino-defined (CDAA) diet (Research Diets, Cat. No A16092201i), high fat diet (Research Diets, Cat. No D12492; 60% fat by calories) or high sucrose diet (Research Diets Cat. No D19022101), as indicated in figure legends. Mouse experiments were performed on age-, sex- and strain-matched littermates. All animal procedures were conducted in compliance with protocols approved by the Regeneron Pharmaceuticals Institutional Animal Care and Use Committee.

## Mice and human genotyping

DNA from mouse and human livers were extracted by DNeasy Blood & Tissue Kit (Qiagen). DNA area including mouse *Marc1* Exon 2 and 3 regions was amplified using Platinum SuperFi II DNA Polymerase (Thermo Fisher) using primers 5′-aattgctgctacctggtgct-3′ and 5′-tggttcatgagggttgtcgg-3′. PCR product was digested by EcoRV restriction enzyme (New England Biolabs) and separated on 0.8% Agarose gel by electrophoresis. Pair of primers 5′-gctaggagcagcttttctga-3′ and 5′-caacagagccgaggtcatca-3′ were used to amplify whole human *MARC1* gene exon 3 by PCR reaction, and ExoSAP-IT Express PCR Product Cleanup Kit (Thermo Scientific) was used to remove both primers before sequencing. Human *MARC1* 165A/T and 187M/K genetic variants were detected by Sanger sequencing.

## Human samples

Fresh frozen human liver samples were acquired from various vendors (LifeNet Health, BioIVT and QPS Medicals). Samples were selected by shortest time period between patient deceased and when liver samples were collected, no RNA degradation (RNA integrity number (RIN) score), absence of hepatitis B and C, liver oncology, severe liver inflammation/fibrosis and minimal or no alcohol use history (all reported by vendor).

## RNA preparation and RNA-sequencing read mapping

Total RNA was purified from Wild type and heterozygous/homozygous Marc1 or Marc2 knockout mice tissues using MagMAX-96 for Microarrays Total RNA Isolation kit (Life Technologies), according to manufacturer's specifications. Genomic DNA was removed using MagMAX Turbo DNase buffer and TURBO DNase from the MagMAX kit listed above. Messenger RNA was purified from total RNA using Dynabeads mRNA purification kit (Invitrogen). Strand-specific RNA sequencing (RNA-seq) libraries were prepared using KAPA mRNA-Seq Library Preparation kit (Kapa Biosystems). After mRNA quality control analysis, RNA-seq libraries were prepared and long read sequencing (75bp, single end, 15M reads) was used for all samples. Sequencing was performed on Illumina HiSeq 2500 by a multiplexed single-read run with 33 cycles. Raw sequence data (BCL files) were converted to FASTQ format via Illumina bcl2fastq v2.17. Reads were decoded based on their barcodes and read quality was evaluated with FastQC (www.bioinformatics.babraham.ac.uk/projects/fastqc/). Reads were mapped to the mouse genome (NCBI GRCm38) using ArrayStudio software (OmicSoft Corp.) allowing two mismatches. Differential expression analysis was performed using DESeq2 (v1.38.2) with default parameters.

## qRT-PCR (Taqman)

Liver and white adipose tissue (~50 mg) or cell line (HuH-7) pellet were homogenized in Trizol (Thermo Fisher), and chloroform was used for phase separation. The aqueous phase, containing total RNA, was purified using MagMAX-96 for Microarrays Total RNA Isolation Kit (Life Technologies), according to manufacturer's instructions. Genomic DNA was removed using RNase-Free DNase Set. SuperScript VILO Master Mix (Thermo Fisher) was then used to reverse-transcribe mRNA into cDNA. The resultant cDNA was amplified with SensiFAST Probe Lo-ROX using the 12K Flex System (BioCat.). Data are presented as a change in gene expression relative to the housekeeping gene, mouse beta-actin (*Actb*) or human *36B4* (*RPLP0*), determined using the delta-delta-Ct method [46]. RT-PCR (Taqman) primer sequences for mouse *Marc1* mRNA detection: probe sequence CCTCCAGTGCAGAGTG-CATGGC, forward primer: GCCTGCCACAAACCCACT and reverse primer

GAGCTGCATCCTCTCCACAATC, targeting mouse *Marc1* mRNA exon 2 and 3 of open reading frame.

## Expression of MARC1 in cultured hepatocytes

The cDNA for human *MARC1* (NCBI NM_022746.4) was cloned downstream of the constitutive HSV thymidine kinase (pTK) [47] or CMV promoter in pcDNA 3.1+ plasmid backbone. Single or triple FLAG tags (DYKDDDDK) with GGGGS linker were placed at the C-terminus of each plasmid construct. In some initial studies pCMV-XL6 (C- terminal 1x FLAG/MYC tag, CMV promoter) plasmids (Origene) were used [5]. Single nucleotide changes were introduced using the QuikChange site-directed mutagenesis kit (Stratagene) and confirmed by Sanger sequencing.

Human hepatoma (HuH-7) cells, which tested negative for mycoplasma, were obtained from the Creative Bioarray (Cat. No CSC-C9441L), transfected with plasmids expressing wild-type or variant human *MARC1* using TransIT-LT1 Transfection Reagent (Mirus). HuH-7 cells were grown to 90% confluence in high glucose DMEM with 10% full calf serum (FCS) plus 100 IU/ml penicillin and 100 μg/ml streptomycin, cells cultivated in 37°C 5% $CO_2$ incubator.

## Assessment of MARC1 protein degradation

Plasmids encoding empty vector (EV), wild-type (WT) and common variants (165T) of human MARC1 (with C terminal 1x FLAG tag, pcDNA 3.1+ plasmid backbone) were transfected in HuH-7 cells using TransIT-LT1 Transfection Reagent (Mirus, Cat. No MIR 2300) according to manufacture protocol. Cells were grown 48 hours in full media (high glucose DMEM with 10% full calf serum (FCS) plus 100 IU/ml penicillin and 100 μg/ml streptomycin), then DMSO only (vehicle) or 10 uM MG-132 in DMSO (proteasomal inhibitor) or 10ug/ml chloroquine in water (lysosomal inhibitor) plus DMSO separately in the same dish were added in cell media, cell incubated for additional 8 hours. Later, cells were collected by scraping, solubilized in 1x RIPA buffer (Millipore) with protease inhibitors, proteins measured, denatured/reduced, and separated on SDS-PAGE (4–20%) gels and blotted with corresponding antibodies Flag tag (MARC1-Flag), LC3I/II, Ubiquitin and Calnexin antibodies. Ubiquitin was used as proteasomal inhibitor control, LC3B-I/II as lysosomal inhibitor control [48]. In the cycloheximide (CHX) chase assay, CHX (Sigma-Aldrich, Cat. No C7698; 300 μg/ml in DMSO) was added dropwise to the full media of HuH-7 cells. The degradation of the MARC1 protein was then assessed at 0, 3, 6, and 9 hour intervals. In addition, a combination treatment of CHX (300 μg/ml) plus MG-132 (10 uM) or CHX (300 μg/ml) plus chloroquine (10ug/ml) was utilized. DMSO was used for the control group treatment and was added to each well with the combinational treatments.

## Western blot

Tissues or hepatic cell line pellets were resuspended in 1x RIPA buffer (EMD Millipore) supplemented with protease inhibitors (Sigma Aldrich) and disrupted by 10 passages through a 25-gauge needle (for cell lines), but tissues were disrupted with FastPrep-24 5G homogenizer with ceramic beads (MP Biomedicals, Cat No 116005500/6913-500)). The homogenates were centrifuged at 12,000 × g for 15 min at 4°C to collect postnuclear fraction. Proteins from white adipose tissues were extracted by commercial kit (Invent Biotechnologies, Cat. No AT-022). Post-nuclear supernatant protein concentrations were determined by BCA Protein Assay (Thermo Scientific, Cat. No 23227). Identical amounts (10–60 μg) of protein were size-fractionated on 4–15 or 4–20% gradient SDS-PAGE gels (Bio-Rad) under reducing conditions, transferred to PVDF membranes (Immobilon P, Millipore, Cat. No IPVH00010). The

membranes were blocked with 5% milk in 0.1% Tween-20 and probed with the indicated primary antibodies, signal detected using an enhanced chemiluminescent detection system (Pierce Biotechnology, Cat. No ECL 37069) and imaged with Amersham Imager 600 (GE Healthcare Life Sciences). Amersham Imager 600 built-in software was used to detect intensities of blot signals. Primary antibodies used in studies: Calnexin (Enzo Life, Cat. No ADI-SPA-860-J; 1:3000), Flag tag (Sigma Aldrich, Cat. No F1804; 1:1000), mouse CYB5B (Proteintech, Cat. No 15469-1-AP; 1:1000), mouse CYB5R3 (Proteintech, Cat. No 10894-1-AP; 1:1000), human MARC1 (Abcepta, Cat. No AP9754c 1:1000), human MARC2 (Proteintech, Cat. No 24782-1-AP 1:1000), mouse Marc2 (Sigma Aldrich, Cat. No HPA015085; 1:1000), Ubiquitin (Cell Signaling, Cat. 3936; 1:1000), LC3-I/II (Cell Signaling, Cat. No 3868, 1:1000).

### Marc1 antibody production and validation

Polyclonal anti-mouse Marc1 antibodies were produced by immunization of rabbits with fragment of mouse Marc1 protein (NCBI NP_001277202.1, aa191-340) produced in E. coli. After the last immunization of rabbits, serum antibodies were affinity purified using mouse Marc1 specific peptide CSEQALYGKLPIFGQYFALENPGTI (aa 306–329; GenScript). Generated antibodies were validated on wild type and *Marc1* KO mouse liver samples using Western Blot. In most experiments purified rabbit anti mouse Marc1 IgG (batch DB0161 (1ug/mL)) was used.

### Liver histology

Liver tissues were collected from isoflurane-anesthetized/cervical dislocated mouse, fixed in 4% paraformaldehyde (PFA) for 24 hours. The fixed tissues were embedded in paraffin, sectioned, and stained with hematoxylin and eosin (Histoserv, Inc.).

Picro-Sirius Red staining was done on PFA fixed, unstained paraffin embedded liver sample slides. After heating, the slides were deparaffinized with xylene, ethanol, and distilled water, then stained in Bouin's solution (Sigma Aldrich Cat. No HT10132) and 0.1% Picro-Sirius Red solution (Sigma Aldrich Cat. No 365548), fixed with 0.5% acetic acid solution. Finally, slides were washed and mounted using Cytoseal (Epredia, Cat. No 831016).

For CD45 immunohistochemistry staining, automated staining was performed on Leica Bond RX autostainer (Leica Biosystems). Slides were deparaffinized with Bond Dewax solution (Leica Biosystems, Cat. No AR9222) followed by antigen retrieval with ER2 (EDTA-based) buffer (Leica Biosystems, Cat. No AR9640) at 100˚C for 30 minutes. Anti-CD45 antibody (Cell Signaling, Cat. No 70257) was added at 0.1 ug/ml followed by 1 hour incubation at room temperature, followed by chromogenic detection with Bond Polymer Refine Detection kit (Leica Biosystems, Cat. No DS9800) that includes anti-rabbit HRP polymer, DAB and hematoxylin counterstain. Slides were washed between steps with Bond wash solution (Leica Biosystems, Cat. No AR9590). After staining, slides were dehydrated in increasing concentrations of ethyl alcohol and xylenes and mounted with Surgipath Micromount mounting medium (Leica Biosystems, Cat. No 3801731), dried and scanned on Aperio AT2 scanners (Leica Biosystems).

To measure area of cytosolic lipid droplets, fibrosis or CD45 positive cells, stained slides were scanned with Leica Aperio AT-2 Scanner and whole tissue area, non-stained area, and the area of vessels were qualified by HALO Classifiers software. The area of lipid droplets/ fibrosis/CD45 (%) was calculated by subtracting the area of vessels from non-stained area and then divided by the whole tissue area.

### Imaging of MARC1 intracellular localization

HuH-7 cells (Creative Bioarray, Cat. No CSC-C9441L) were cultured in complete medium (high glucose Dulbecco's modified Eagle's medium with 10% fetal calf serum (FCS) plus 100

IU/ mL penicillin and 100 mg/mL streptomycin. On day 0, cells were transfected with pcDNA 3.1+ (CMV promoter) containing plasmids with no gene (empty vector) or human MARC1 wild type and common variants with C terminal FLAG/MYC tags. Transfection was done with Neon Transfection System (Thermo Fisher, Cat. No MPK5000/MPK1025) with pre-validated settings (pulse– 1230 V; pulse width– 20 ms; pulse numbers: 3) according to producer instructions. Cells were plated on μ-Slide dishes covered with ibiTreat (Ibidi, Cat. No 80826) in full media for 48 hours, after cell media was changed to pre-warmed serum-free media with 200nM Mitotracker Orange CMTMRos staining solution (Thermo Fisher, Cat. No M7510) and incubated for 20 minutes in 37˚C 5% $CO_2$ in the dark. Cells were washed, fixed and permeabilized with Triton X100 and blocked with 5% donkey serum supplemented by 2% BSA. After the final wash, cells were incubated with Flag antibody (Sigma Aldrich, Cat. No F1804; 1:1000) for 1h RT, washed and incubated with secondary antibody (Alexa Fluor 647 AffiniPure Donkey Anti-Mouse IgG (Jackson ImmunoResearch Inc., Cat. No 715-605-151) RT for 1 hour. After the final wash, cells were incubated in Mounting Medium with DAPI (Ibidi, Cat. No 50011) and imaged with Leica confocal microscope.

## Marc1/Marc2 benzamidoxime (BAO) reduction assay in mouse primary hepatocytes

10–12 week old male Cas9 transgenic mice (Cas9-Ready V2.5) were injected in tail vein with $3 \times 10^{10}$ genome copies (GC) of AAV8 containing Cas9 gRNA against *Marc1*, *Marc2*, *Marc1/2* (one gRNA targeting both genes) or control gRNA (anti red fluorescent protein (*RGF*)). 2–4 weeks after virus injection, mice were sacrificed by lethal isoflurane inhalation and cervical dislocation, and primary hepatocytes isolated, counted and plated on poly-D-lysine pre-coated plates [49]. Cells were kept in incubator (37˚C, 5% $CO_2$) in full media (Williams E media with 10% full calf serum (FCS) plus 100 IU/ml penicillin, 100 μg/ml streptomycin and 2 mM L-glutamine) to attach to the plate (4h). After attachment, cells were washed with PBS once and incubated in serum-free media (low glucose DMEM plus 100 IU/ml penicillin, 100 μg/ml streptomycin and 2 mM L-glutamine, 10 μM sodium molybdate dihydrate) with 0.5 mM benzamidoxime for 2 hours. After reaction, cell media was collected, spun 1000 x G for 5 min at 4˚C, supernatant collected, reaction stopped by adding ultra-pure methanol (1:1, vol/vol), spun 21000 x G 5 min 4˚C, supernatant collected and diluted in 20 mM ammonium bicarbonate 1:5 (vol/vol). After media collection, protein from all primary cell plates were extracted by 1x RIPA buffer with protease inhibitors, total protein amount measured by BCA assay. Reduced protein lysates in Laemmli SDS sample buffer were size-fractionated on 4–20% SDS-PAGE gel and blotted with corresponding antibodies.

Benzamidoxime (BAO) and its metabolite benzamidine (BA) were measured in cell media by HPLC-UV. All HPLC analyses were performed using a Waters ACQUITY I-Class UPLC system Separation Module with a Waters ACQUITY UPLC Photodiode Array Detector. Waters Empower 3 Build 3471 integration software was used for data analysis. Benzamidoxime and benzamidine were separated on a Waters Viridis CSH Fluoro-Phenyl Column with a particle size of 1.7 μm, a diameter of 2.1 mm, and a length of 50 mm. The mobile phase was composed of 20 mM ammonium bicarbonate at pH 8.5–8.6 and 30% acetonitrile (vol/vol) under isocratic conditions with a flow rate of 0.2 mL/min and a sample injection volume of 2 μL. The total run time was 5 minutes (retention time of benzamidoxime: 1.1 ± 0.2 minute, benzamidine: 1.7±0.2 minute), and the detection wavelength was set at 229 nm. The column temperature was maintained at 40˚C, while the sample storage temperature was kept at 5˚C. To quantify Benzamidoxime depletion and Benzamidine conversion, calibration curves were used with concentration ranges of 0.1–2 mM and 0.00832–1.04 mM, respectively. In addition, a

diluted cell media blank was injected to check for any potential interference in both benzamidoxime and benzamidine measurements. All standards were prepared in experiment used cell media to closely mimic the sample matrix.

## Lipid analysis

Circulating triglycerides (TG), total cholesterol (TC), LDL-C, non-esterified fatty acids (NEFA), alanine aminotransferase (ALT), aspartate aminotransferase (AST), urea nitrogen, creatinine, glucose, inorganic phosphate and alkaline phosphatase (ALK) levels were determined in serum using ADVIA Chemistry XPT blood chemistry analyzer (Bayer). Non-HDL-C levels were calculated by subtracting HDL-C from TC values. Liver lipid levels were evaluated as previously described [39,50].

## Statistical analysis

Statistical and graphical data analyses were performed using Microsoft Excel and Prism 9 (GraphPad Software, Inc.). Mean values were compared using unpaired two-tailed t-tests or one-way ANOVA. Data shown as mean values with standard error. In box and whisker plots, the middle line is plotted at the median, the upper and lower hinges correspond to the first and third quartiles, and the upper and lower whiskers display the full range of variation (minimum to maximum). Grubbs' test was used to determine and remove significant outliers.

## Supporting information

**S1 Fig. MARC1 pLOF variants are associated with lower plasma ALT levels.** Rare variant adjusted by common variants. ALT (U/L); AAF—alternate allele frequency.
(TIF)

**S2 Fig. MARC1 pLOF variants affect plasma lipids and Apoprotein B levels but have no effect on coronary artery disease or type 2 diabetes.** (A) Serum lipids, glucose, body mass index and blood pressure, (B) type 2 diabetes and coronary artery disease.
(TIF)

**S3 Fig. MARC2 pLOF coding variants have no effect on plasma ALT levels.** AAF—alternate allele frequency.
(TIF)

**S4 Fig. MARC1 A165T substitution leads to reduction in MARC1 protein expression in HepG2 cells.** (A) Plasmids with empty vector (Vector), wild-type and common variants of human MARC1 (CMV promoter, C terminal 1x Flag tag) were expressed in HepG2 cells. 48 hours after transfection, cells were split into two parts. First part of cells were solubilized in RIPA buffer, proteins separated on SDS-PAGE (4–20%) gels and blotted with corresponding antibodies, (B) immunoblot quantitation was performed. (C) Second part of cells was used to measure MARC1 mRNA levels using Real-Time PCR. Mean ± s.e.m. are shown in all graphs, ****p<0.0001
(TIF)

**S5 Fig. Human MARC1 p.165T variant causes mislocalization of MARC1 protein in hepatic cells.** HuH-7 cells were transfected with plasmids containing wild type or 2 common genetic variants (p.A165T and p.M187K) of human MARC1 (with C terminal Flag tag and gene expression driven under CVM promoter). 48 hours after transfection cells were incubated with Mitotracker Orange CMTMRos staining (Thermo Fisher) solution in serum free media for 20 minutes, cells were washed, fixed, permeabilized and blocked as described in

Methods. After additional washing, cells were incubated with Flag antibody (Sigma) for 1 hour, washed and incubated with Alexa Fluor 647 conjugated secondary antibody. After final wash, cells were incubated in mounting medium with DAPI (Ibidi) and imaged with Leica confocal microscope. The experiment was repeated once with the equivalent results.
(TIF)

**S6 Fig. Accelerated degradation of human MARC1 p.165T variant is not mediated by proteasomal or lysosomal pathways in HuH-7 cells.** (A) Plasmids encoding empty vector (EV), wild-type (WT) and common variants (165T) of full length human MARC1 (with C terminal FLAG tag) were transfected into HuH-7 cells. 48 hours after transfection, cell media containing DMSO only (vehicle) or 10 μM MG-132 in DMSO (proteasomal inhibitor) or 10ug/ml chloroquine in water (lysosomal inhibitor) plus DMSO separately were added and cells were incubated for additional 8 hours. After that, cells were washed, collected and solubilized in RIPA buffer, proteins separated on SDS-PAGE (4–20%) gels and blotted with corresponding primary antibodies. (B) Protein densitometry was performed on blots (A) and ratio between MARC1 variants and Calnexin were calculated. (C) Cells were prepared and transfected with WT or A165T variant human MARC1 similar as in (A), 48 h after transfection cells were incubated with DMSO or cycloheximide (CHX; 300 μg/ml) in DMSO or CHX (300 μg/ml) plus 10 μM MG-132 in DMSO or CHX (300 μg/ml) plus 10ug/ml chloroquine in DMSO for 0, 3, 6 and 9 hours. After incubation cells were collected and processed as in (A). (D) Protein densitometry was performed on (C) blots and ratios between MARC1 variants and β-actin were calculated, the value of 0 timepoint for each treatment condition were set as 1 and the relative changes in the protein levels were calculated over time. Ubiquitin was used as proteasomal inhibitor control, LC3B-I/II as lysosomal inhibitor control. CANX (calnexin) and β-actin–as protein loading controls. The experiments was repeated once with the same results.
(TIF)

**S7 Fig. Marc1 gene deletion has no effect on circulating or liver lipids in male mice fed choline-deficient L-amino acid defined high fat diet (CDAA-HFD).** Nine weeks old male mice (11 WT and 10 Marc1 KO) were fed with CDAA-HFD for 18 weeks. (A) Body mass gain was measured every 2 weeks, (B) body composition (Micro CT) of mice were assessed at week 6 of experiment. (C) Serum, (D) liver lipids, (E) selected liver gene expression (mRNA by qRT-PCR) were performed at end of experiment. Each point represents individual mouse, mean ± s.e.m. are shown in all graphs, *p<0.05, ***p<0.001, ****p<0.0001.
(TIF)

**S8 Fig. Marc1 gene deletion has no effect on liver lipid accumulation, inflammation or fibrosis in mice fed choline-deficient L-amino acid defined high fat diet (CDAA-HFD).** (A) Liver samples were collected from mice described in S7 Fig at end of the experiment. Fixed liver sections were stained with hematoxylin/eosin (HE), CD45 and Sirius Red staining. (B) Lipid droplet area, (C) CD45 positive cell area and (D) fibrosis area were measured from each animal liver section. Slides were scanned on Aperio AT2 scanners (Leica Biosystems) with 10X magnification. Each point represents individual mouse, mean ± s.e.m. are shown in all graphs.
(TIF)

**S9 Fig. Marc1 gene deletion has no effect on body weight or composition in female mice fed high fat high fructose diet (HFHDF).** (A) 7–10 weeks old female mice were fed with HFHFD for 12 weeks. Body weight was measured every 2 weeks, (B) body composition (MRI) in mice were assessed at week 13 of experiment. Each point represents individual mouse, mean ± s.e.m. are shown in all graphs
(TIF)

**S10 Fig. Marc2 gene deletion in mouse affects body mass composition.** Wild type and Marc2 homozygous female mice (n = 5–9), fed chow ad lib body mass composition (Micro CT) were estimated at age 6–7 weeks. Each point represents individual mouse, mean ± s.e.m. are shown in all graphs, \*\*\*p<0.001, \*\*\*\*p<0.0001.
(TIF)

**S1 Table. Rare, novel predicted loss of function variants in MARC1 gene detected in this study.** Hgvsc—Human Genome Variation Society DNA variant abbreviation, Hgvsp—Human Genome Variation Society coding protein abbreviation, Met1?–skip of first Methionine of protein.
(TIF)

**S2 Table. MARC1 pLOF variants show association with reduced liver fat and protection from various etiology liver diseases.** AAF—alternate allele frequency, PDFF-MRI—proton density fat fraction by magnetic resonance imaging.
(TIF)

**S3 Table. MARC1 pLOF variants are associated with changes in plasma lipids but show no effect on cardiometabolic traits and other metabolic measurements.** AAF—alternate allele frequency.
(TIF)

**S4 Table. Marc1 gene deletion does not affected Mendelian patterns of allele inheritance in mice.** Genotypes of pups from Marc1 heterozygous knockout mice breeding (F2-F4) were detected, results grouped by sex and genotype.
(TIF)

**S5 Table. Feeding Marc1 KO male mice with different NASH/NAFLD-causing diets revealed minimal difference in liver lipids and plasma metabolites compared to Marc1 WT littermates.** Mice fed with chow diet were sacrificed at age 13 weeks. For other diets—8–10 weeks old WT and Marc1−/− male mice were fed with HFHFD for 35 weeks or CDAA-HFD for 18 weeks or high fat diet (HFD) for 11 weeks or high sucrose diet (HSD) for 9 weeks, respectively. At the end of experiments mice were sacrificed, liver and serum collected and indicated measurements done as described in Methods. Each group contains 7–12 mice. All mice were sacrificed after *ad lib* feeding state (non-fasted). Mean ± s.e.m. are shown for all measurements. In bold–all measurement with \*p<0.05.
(TIF)

**S6 Table. Feeding Marc1 KO female mice with different NASH/NAFLD-causing diets revealed no changes in liver lipids and plasma metabolites compared to Marc1 WT littermates.** Serum was collected from chow diet fed mice at age 7–9 weeks. For other diets, 7–10 weeks old wild-type and Marc1−/− mice were fed with HFHFD or CDAA-HFD for 12 weeks, respectively. At the end of experiments mice were sacrificed, liver and serum collected and indicated measurements performed as described in Methods. All mice were sacrificed at *ad lib* feeding state (non-fasted). Each group contains 10–11 mice. Mean ± s.e.m. are shown for all measurements. NA–not measured.
(TIF)

**S7 Table. Minimal changes in liver gene expression profile were observed after whole body Marc1 gene ablation.** RNAseq was performed using livers from ad lib chow fed WT and Marc1 KO male mice (n = 8–9). Fold changes >1.5X between Marc1 KO vs WT mice are

shown. P-value calculated by t-test, adjusted p-values corrected by multiple variants. WT gene expression–mean values of mRNA expression (TPM) of wild type mice.
(TIF)

**S8 Table. RNAseq analysis of mouse tissues mRNA expression profile for WT, Marc1 KO and Marc2 KO mice.** mRNA was extracted from 6–10 weeks old WT, Marc1 or Marc2 KO male mice, fed ad lib with chow diet. RNA extraction, processing and analysis described in details in Supplemental Methods.
(XLSX)

**S9 Table. Reagents listed in the study.**
(TIF)

**S1 Appendix. Numerical data for 2, 4A, 4B, 5C, 6–8; S4, S6–S10 Figs and S5, S6 Tables.**
(XLSX)

## Acknowledgments

We thank Erqian Na, Corey Alexa-Braun, Helen Chen, Shunhai Wang, Ning Li, Lampros Panagis, Jarrell Wiley and David D'Ambrosio for excellent technical assistance. We thank Regeneron Genetics Center for providing intellectual and technical support for this study.

We thank the UK Biobank team, their funders, the dedicated professionals from the member institutions who contributed to and supported this work, and the UK Biobank participants. The exome sequencing was funded by the UK Biobank Exome Sequencing Consortium (i.e., Bristol Myers Squibb, Regeneron, Biogen, Takeda, AbbVie, Alnylam, AstraZeneca and Pfizer). This research has been conducted using the UK Biobank Resource under application number 26041. We thank the MyCode Community Health Initiative participants for taking part in the DiscovEHR collaboration. We acknowledge the Penn Medicine BioBank (PMBB) for providing data and thank the patient-participants of Penn Medicine who consented to participate in this research program. The PMBB is approved under IRB protocol# 813913 and supported by Perelman School of Medicine at University of Pennsylvania, a gift from the Smilow family, and the National Center for Advancing Translational Sciences of the National Institutes of Health under CTSA award number UL1TR001878. Furthermore, the authors would like to acknowledge the support from Lund University Infrastructure grant"Malmö population-based cohorts" (STYR 2019/2046). We acknowledge the BioMe team for providing data and thank the patient-participants of Mt. Sinai BioMe Biobank who consented to participate in this research program. The BioMe healthcare delivery cohort at Mount Sinai was established and maintained with a generous gift from the Andrea and Charles Bronfman Philanthropies.

The Genotype-Tissue Expression (GTEx) project was supported by the Common Fund of the Office of the Director of the National Institutes of Health, and by NCI, NHGRI, NHLBI, NIDA, NIMH, and NINDS. Data obtained from: the GTEx Portal with dbGaP accession number phs000424.v8.p2 on 11/29/2022.

## Author Contributions

**Conceptualization:** Luca A. Lotta, Andrew J. Murphy, Mark W. Sleeman, Viktoria Gusarova.

**Data curation:** Eriks Smagris, Niek Verweij, Gabor Halasz.

**Formal analysis:** Eriks Smagris, Lisa M. Shihanian, Ivory J. Mintah, Yuliya Livson, Niek Verweij, Gabor Halasz.

**Investigation:** Eriks Smagris, Lisa M. Shihanian, Ivory J. Mintah, Parnian Bigdelou, Yuliya Livson, Heather Brown, Niek Verweij.

**Methodology:** Eriks Smagris, Yuliya Livson, Niek Verweij, Viktoria Gusarova.

**Project administration:** Eriks Smagris, Niek Verweij, Viktoria Gusarova.

**Supervision:** Charleen Hunt, Reid O'Brien Johnson, Suzanne A. Hartford, Luca A. Lotta, Viktoria Gusarova.

**Validation:** Eriks Smagris, Lisa M. Shihanian, Ivory J. Mintah, Yuliya Livson, Niek Verweij.

**Writing – original draft:** Eriks Smagris, Niek Verweij, Viktoria Gusarova.

**Writing – review & editing:** Eriks Smagris, Niek Verweij, Charleen Hunt, Reid O'Brien Johnson, Tyler J. Greer, Suzanne A. Hartford, George Hindy, Luanluan Sun, Jonas B. Nielsen, Luca A. Lotta, Andrew J. Murphy, Mark W. Sleeman, Viktoria Gusarova.

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
