## [Decision Letter · Decision Letter 0]

21 Nov 2023

Dear Dr Gusarova,

Thank you very much for submitting your Research Article entitled 'Divergent role of Mitochondrial Amidoxime Reducing Component 1 (MARC1) in human and mouse' to PLOS Genetics.

The manuscript was fully evaluated at the editorial level and by independent peer reviewers. The reviewers appreciated the attention to an important problem, but raised some substantial concerns about the current manuscript. Based on the reviews, we will not be able to accept this version of the manuscript, but we would be willing to review a much-revised version. We cannot, of course, promise publication at that time.

If you decide to revise the manuscript for further consideration at PLOS Genetics, please aim to resubmit within the next 60 days, unless it will take extra time to address the concerns of the reviewers, in which case we would appreciate an expected resubmission date by email to plosgenetics@plos.org.

We are sorry that we cannot be more positive about your manuscript at this stage. Please do not hesitate to contact us if you have any concerns or questions.

Yours sincerely,

Jiandie Lin

Academic Editor

PLOS Genetics

Hua Tang

Section Editor

PLOS Genetics

Reviewer's Responses to Questions

**Comments to the Authors:**

Reviewer #1: Smagris and colleagues have examined the linkage between variations in MARC1 and its protein stability and have then gone on to evaluate the effects of MARC1 deficiency in mice. Variation in the gene encoding MARC1 have been linked to hepatic steatosis in people but a mechanistic understanding of how altered MARC1 activity affects liver lipid content is lacking. This work is therefore regarded as of high importance. The studies are well done and the experimental progression is very logical. There were a few remaining questions that are fairly minor.

As proof of concept, it would be useful to complement the Marc1/2 KO hepatocytes by expressing the human and mouse MARC1 protein to see if there are species differences in enzyme activity.

To distinguish between impaired stability and translation of the MARC1 variant protein, studies with cyclohexamide would be useful.

What is the expression pattern of MARC2 in humans?

Minor:

Last paragraph of the results: Mice don’t have “genders”. This is a human-specific concept. This should also be fixed in the methods

Reviewer #2: In their exome-wide association study, Eriks et al uncovered several novel loss of function variants in MARC1. They conducted studies to elucidate the protein stability and localization of well-known A165T and M187K mutations. Their finding revealed that the A165T mutation results in lower protein stability compared to the wild type. They also generated the Marc1 deficient mice which exhibited no significant differences in NASH progression. The generated Marc2 mice displayed neurological behavior defects. They also found Marc2 as a key enzyme involved in N-hydroxylated amidine reduction in mouse liver.

Despite the presence of known data and numerous negative findings, the creation of Marc1 knockout mice and the development of a specific Marc1 antibody hold promise for advancing the field. The authors propose that their study serves as an example of the non-consistency between human and mouse models, emphasizing the importance of investigating such discrepancies in research outcome.

Major questions:

1: In Figure 8, the AAV-mediated knockdown of Marc1 and Marc2 was exclusively conducted on hepatocytes. It would be valuable to know if these knockdown mice were subjected to a NASH-inducing diet. Additionally, exploring the effects of overexpressing the A165T and M187K mutants in Marc1 knockdown mouse livers could provide interesting insights.

2: Have the authors investigated the impact of a normal high-fat diet (HFD) on Marc1-deficient mice

3: In Figure S4, the protein levels for A165T are noticeably higher in MG-132 and Chloroquine compared to DMSO. This raises questions about the authors' claim that the A165T variant was not rescued by these inhibitors. Is there any existing literature reporting the degradation of wild-type Marc1 through the proteasomal or lysosomal pathways. Further investigation into protein degradation using CHX treatment will be interesting.

4: The image quality in Figure 3 is suboptimal, particularly in discerning whether wild-type MARC1 localizes with mitochondria.

5: In Figure 4A-B, the statement "we found Marc2 mRNA expression was almost two times higher than Marc1 in mouse liver" may be challenging to interpret due to the inherent difficulties in comparing expression levels between two different genes.

**Have all data underlying the figures and results presented in the manuscript been provided?**

Reviewer #1: Yes

Reviewer #2: Yes

PLOS authors have the option to publish the peer review history of their article (what does this mean?). If published, this will include your full peer review and any attached files.

Reviewer #1: No

Reviewer #2: No

---

## [Decision Letter · Decision Letter 1]

9 Feb 2024

Dear Dr Gusarova,

We are pleased to inform you that your manuscript entitled "Divergent role of Mitochondrial Amidoxime Reducing Component 1 (MARC1) in human and mouse" has been editorially accepted for publication in PLOS Genetics. Congratulations!

Yours sincerely,

Jiandie Lin

Academic Editor

PLOS Genetics

Hua Tang

Section Editor

PLOS Genetics

Comments from the reviewers (if applicable):

Reviewer's Responses to Questions

**Comments to the Authors:**

Reviewer #1: Thank you for addressing some of the prior concerns.

Reviewer #2: The authors have addressed all the questions.

**Have all data underlying the figures and results presented in the manuscript been provided?**

Reviewer #1: Yes

Reviewer #2: None

PLOS authors have the option to publish the peer review history of their article (what does this mean?). If published, this will include your full peer review and any attached files.

Reviewer #1: No

Reviewer #2: **Yes: **Linkang Zhou

**Data Deposition**

http://datadryad.org/submit?journalID=pgenetics&manu=PGENETICS-D-23-01107R1

**Press Queries**

---

## [Editor Report · Acceptance letter]

25 Feb 2024

PGENETICS-D-23-01107R1 

Divergent role of Mitochondrial Amidoxime Reducing Component 1 (MARC1) in human and mouse 

Dear Dr Gusarova, 

We are pleased to inform you that your manuscript entitled "Divergent role of Mitochondrial Amidoxime Reducing Component 1 (MARC1) in human and mouse" has been formally accepted for publication in PLOS Genetics! Your manuscript is now with our production department and you will be notified of the publication date in due course.

With kind regards,

Zsofia Freund

PLOS Genetics

On behalf of:
